# Membership Inference Attacks and Privacy in Topic Modeling

**Nico Manzonelli**                                                            *nicomanzonelli@gmail.com*
*John A. Paulson School of Engineering and Applied Sciences*
*Harvard University*

**Wanrong Zhang**
*John A. Paulson School of Engineering and Applied Sciences*
*Harvard University*

**Salil Vadhan**
*John A. Paulson School of Engineering and Applied Sciences*
*Harvard University*

**Reviewed on OpenReview:** *https://openreview.net/forum?id=NmWp5lFL7L*

## Abstract

Recent research shows that large language models are susceptible to privacy attacks that infer aspects of the training data. However, it is unclear if simpler generative models, like topic models, share similar vulnerabilities. In this work, we propose an attack against topic models that can confidently identify members of the training data in Latent Dirichlet Allocation. Our results suggest that the privacy risks associated with generative modeling are not restricted to large neural models. Additionally, to mitigate these vulnerabilities, we explore differentially private (DP) topic modeling. We propose a framework for private topic modeling that incorporates DP vocabulary selection as a pre-processing step, and show that it improves privacy while having limited effects on practical utility.

## 1 Introduction

Deep learning models' propensity to memorize training data presents many privacy concerns. Notably, large language models are particularly susceptible to privacy attacks that exploit memorization (Carlini et al., 2021; 2023). In this paper, we aim to investigate whether simpler probabilistic models, such as topic models, raise similar concerns. Our specific focus lies on examining probabilistic topic models like Latent Dirichlet Allocation (LDA) (Blei et al., 2003).

Topic modeling is an unsupervised machine learning (ML) method that aims to identify underlying themes or topics within a corpus. Despite the recent successes of large language models (LLMs), Probabilistic topic models are still widely used due to their interpretability and straightforward implementation for text analysis. Furthermore, topic models continue to be developed for a variety of applications and downstream tasks like document classification, summarization, or generation (Boyd-Graber et al., 2017; Wang et al., 2019).

Researchers apply topic models across a variety of domains where privacy concerns arise. For example, many studies in the medical domain use topic models to analyze datasets with sensitive attributes (Mustakim et al., 2021). Additionally, topic models are widely used for various government or national defense applications. For instance, researchers applied LDA to better understand Twitter activity aimed at discrediting NATO during the Trident Juncture Exercises in 2018 (Uyheng et al., 2020). To ensure ethical and responsible use of ML, it is crucial to consider the privacy implications associated with topic modeling.

Probabilistic topic models like LDA serve as a basic Bayesian generative model for text. Neural topic models and language models can learn complex generative processes for text based on observed patterns in the

training data. However, LDA requires the generative process for documents to be explicitly defined *a priori* with far fewer parameters and a Bag-of-Words (BoW) text representation. Investigating LDA's susceptibility to privacy attacks provides broader context for the privacy vulnerabilities researched in more complex ML models. Additionally, our work informs practitioners who may opt to use topic models under the false impression that they do not share the same vulnerabilities of LLMs.

To explore the privacy in topic modeling, we conduct membership inference attacks (MIAs) which infer whether or not a specific document was used to train LDA. We propose an attack based on an LDA-specific query statistic designed to exploit memorization. This query statistic is integrated into the Likelihood Ratio Attack (LiRA) framework introduced by Carlini et al. (2022). We show that our attack can confidently infer the membership of documents included in the training data of LDA which indicates that the privacy risks in generative modeling are not restricted to large neural models.

To mitigate such vulnerabilities, we explore differential privacy (DP) for topic modeling. DP acts a direct defense against MIAs by providing a statistical guarantee that the output of some data analysis is indistinguishable regardless of any one user's inclusion in the analysis. While several DP topic modeling algorithms in the literature attempt to protect individual privacy, previous works largely disregard the privacy of the model's accompanying vocabulary set (Zhao et al., 2021; Huang & Chen, 2021; Huang et al., 2022; Decarolis et al., 2020; Park et al., 2016; Wang et al., 2022). Instead, they consider the vocabulary set as given or public information. However, in practice, the vocabulary set is derived from the training data and can leak sensitive information. We propose an algorithm for DP topic modeling that provides privacy in both the vocabulary selection and learning method. By incorporating DP vocabulary selection into the private topic modeling workflow, our algorithm enhances privacy guarantees and bolsters defenses against the LiRA, while having minimal impact on practical utility.

## 2 Background and Notation

To begin, let us establish notation and provide a formal definition of topic models.

**Definition 2.1 (Topic Model)** For a vocabulary size of $V$ and $k$ topics, a topic model $\Phi \in [0,1]^{k \times V}$ is a matrix whose rows sum to 1 representing each topic as a distribution over words.

We let $f_{\mathcal{G}}(D)$ denote the process of learning a model parameterized by latent variables in $\mathcal{G}$ on a corpus $D$ with $M$ documents drawn from the underlying data distribution $\mathbb{D}$. The function $f_{\mathcal{G}}$ incorporates the learning algorithm to estimate $\mathcal{G}$'s latent variables and returns the learned topic model $\Phi$. Our definition assumes that the primary objective of topic modeling is to estimate $\Phi$. While $\mathcal{G}$ is also associated with other latent variables that may define the document generation process, $\Phi$ best achieves the goals associated with topic modeling by summarizing the relevant themes in the corpus.

### 2.1 Latent Dirichlet Allocation

Blei et al. (2003) define LDA where each document is assumed to contain a mixture of topics. LDA assumes that each document $d \in D$ is represented by a $k$-dimensional document-topic distribution $\theta \sim \text{Dirichlet}(\alpha)$, and the entire corpus is represented by $k$ $V$-dimensional topic-word distributions $\phi \sim \text{Dirichlet}(\beta)$. Furthermore, it follows a simple bag-of-words approach where each word in a document is generated by first sampling a topic from the document's topic distribution and then sampling a word from the chosen topic's distribution over words.

The process of estimating $\Phi$ presents a Bayesian inference problem typically solved using collapsed Gibbs sampling or variational inference (Griffiths & Steyvers, 2004; Hoffman et al., 2010). Each entry $\Phi_{z,w}$ represents the probability of drawing word $w$ from topic $z$. The entire topic model $\Phi$ represents the likelihood of each word appearing in each topic, which can be used to estimate the document-topic distribution $\theta$ for any given document.

## 2.2 Memorization and Privacy

Understanding memorization in machine learning is critical for trustworthy deployment. Neural model's large parameter space and learning method introduce training data memorization which increases with model size or observation duplication (Carlini et al., 2023; Feldman & Zhang, 2020). Memorization introduces vulnerabilities to attacks on privacy, such as membership inference, attribute inference or data extraction attacks (Shokri et al., 2017; Song & Raghunathan, 2020; Carlini et al., 2021).

Schofield et al. (2017) note that document duplication in topic models concentrates the duplicated document's topic distribution $\theta$ over less topics, and briefly refer to this phenomenon as memorization. Their findings are consistent with studies on memorization and text duplication in large language models (Carlini et al., 2023). In this work, we investigate the memorization and privacy of topic models using membership inference attacks (MIA).

## 2.3 Membership Inference Attacks

In an MIA the adversary learns if a specific observation appeared in the model's training data (Homer et al., 2008; Shokri et al., 2017). These attacks serve as the foundation for other attacks like data extraction, and could violate privacy alone if the adversary learns that an individual contributed to a dataset with a sensitive global attribute (i.e. a dataset containing only patients with disease X).

Prior to attacking ML models, researchers explored exploiting aggregate statistics released in genome-wide association studies (GWAS) (Homer et al., 2008). Shokri et al. (2017) introduced one of the first MIAs on ML models. While most MIAs exploit deep learning models, few studies investigate topic models. Huang et al. (2022) propose three simple MIAs to evaluate their privatized LDA learning algorithm. However, they evaluate their attacks using precision and recall and fail to show that their attack confidently identifies members of the training data.

To evaluate an MIA's ability to confidently infer membership, we examine the attack's true positive rate (TPR) at low false positive rates (FPR) (Carlini et al., 2022). Receiver operator characteristic (ROC) and area under the curve (AUC) analysis may be useful, but we must be wary of the axes. To accurately capture the TPR at the low FPRs of interest, we visualize ROC curves using log-scaled axes.

# 3 MIAs Against Topic Models

In this section, we detail our MIA against topic models based on the LiRA framework, and empirically demonstrate its effectiveness.

## 3.1 Threat Model

We consider a threat model where the adversary has black-box access to the learned topic-word distribution $\Phi$, but not other latent variables captured by $\mathcal{G}$. The adversary can not observe intermediate word counts, learning method, or hyper-parameters used while learning $\Phi$. We also assume that the adversary has query access to the underlying data distribution $\mathbb{D}$, which allows them to train shadow models on datasets drawn from $\mathbb{D}$.

Under the typical query model, we could consider queries as a request to predict a document-topic distribution $\hat{\theta}$ given a document. However, $\hat{\theta}$ would hold little meaning without access to $\Phi$. Furthermore, a clever adversary could query many specially selected documents to reconstruct $\Phi$. Therefore, we remove this level of abstraction and assume that the adversary is given direct access to $\Phi$ as the output of our topic model.

## 3.2 Attack Framework

In the LiRA framework of Carlini et al. (2022), the adversary performs hypothesis testing to determine whether or not a target document $d$ was in the training data. Specifically, we consider $\mathbf{T}_{in}(d) = \{\Phi \leftarrow f_{\mathcal{G}}(D \cup \{d\}) | D \in \mathbb{D}\}$ and $\mathbf{T}_{out}(d) = \{\Phi \leftarrow f_{\mathcal{G}}(D \backslash \{d\}) | D \in \mathbb{D}\}$ where $\mathbf{T}_{in}(d)$ is the distribution of $\Phi$ learned

with $d$, and $\mathbf{T}_{out}(d)$ is the distribution of $\Phi$ learned without $d$. Given an observed $\Phi_{obs}$, we would like to construct the likelihood ratio test statistic $\Lambda$ as follows:

$$\Lambda(\Phi, d) = \frac{p(\Phi_{obs}|\mathbf{T}_{in}(d))}{p(\Phi_{obs}|\mathbf{T}_{out}(d))}, \tag{1}$$

where $p(\Phi_{obs}|\mathbf{T}_x(d))$ is the probability density function of $\Phi_{obs}$ under $\mathbf{T}_x(d)$ (Carlini et al., 2022).

However, the ratio in Equation 1 is intractable because it requires integrating over all $\Phi$ with all possible $D$ with and without $d$. Instead, the adversary applies a carefully chosen *query statistic* $\zeta : (\Phi, d) \to \mathbb{R}$ on all $\Phi$ to reduce the problem to 1-dimension. The test can proceed by calculating the probability density of $\zeta(\Phi_{obs}, d)$ under estimated normal distributions from $\tilde{\mathbf{T}}_{in}(d) = \{\zeta(\Phi, d) \mid \Phi \leftarrow f_{\mathcal{G}}(D \cup \{d\}), D \leftarrow \mathbb{D}\}$ and $\tilde{\mathbf{T}}_{out}(d) = \{\zeta(\Phi, d) \mid \Phi \leftarrow f_{\mathcal{G}}(D \backslash \{d\}), D \leftarrow \mathbb{D}\}$:

$$\Lambda(\Phi, d) = \frac{p(\zeta(\Phi_{obs}, d)|\mathcal{N}(\mu_{in}, \sigma_{in}^2))}{p(\zeta(\Phi_{obs}, d)|\mathcal{N}(\mu_{out}, \sigma_{out}^2))}, \tag{2}$$

where $\mu$ and $\sigma^2$ are the mean and variance of $\tilde{\mathbf{T}}_{in}(d)$ and $\tilde{\mathbf{T}}_{out}(d)$.

In the online LiRA, the adversary must train $N$ shadow topic models with and without $d$ to estimate $\tilde{\mathbf{T}}_{in}(d)$ and $\tilde{\mathbf{T}}_{out}(d)$. Appendix A contains the full online LiRA algorithm. To ease the computational burden of training $N$ shadow topic models for each target document, Carlini et al. (2022) propose an offline variant of the LiRA. In the offline LiRA, the adversary can evaluate any $d$ after learning a collection of $N$ shadow topic models once by comparing $\zeta(\Phi_{obs}, d)$ to a normal estimated from $\tilde{\mathbf{T}}_{out}(d)$ with a one-sided hypothesis test. Appendix B contains the full offline LiRA algorithm.

Designing $\zeta$ is crucial to the attack's success. In supervised learning scenarios, the adversary can directly query the model and use the loss of an observation to inform the statistic (Carlini et al., 2022). However, because topic models present an unsupervised learning tasks, the adversary encounters the challenge of identifying an informative statistic that can be computed efficiently using $\Phi$. Hence, $\zeta$ must be carefully tailored to topic models for the attack to be effective.

### 3.3   Designing an Effective Query Statistic $\zeta$

Effective model query statistics for the LiRA framework must satisfy the following criteria: the statistic should increase for $d$ when $d$ is included in the training data ($\tilde{\mathbf{T}}_{in}(d) > \tilde{\mathbf{T}}_{out}(d)$) to enable the offline LiRA, and the estimated distributions $\tilde{\mathbf{T}}_{in}(d)$ and $\tilde{\mathbf{T}}_{out}(d)$ are approximately normal to allow for parametric modeling. Ideally, the statistic should be related to how the model may memorize the training data to provide an intuitive interpretation for attack performance.

First, we consider statistics from the simple MIAs on LDA presented by Huang et al. (2022). They propose three statistics derived from the target document's estimated topic distribution $\hat{\theta}$: the entropy of $\hat{\theta}$, the standard deviation of $\hat{\theta}$, and the maximum value in $\hat{\theta}$. The intuition behind their chosen statistics comes from the observation that a document's topic distribution tends to concentrate in a few topics when included (or duplicated) in the training data (Schofield et al., 2017). However, Huang et al. (2022) apply global thresholds to these statistics instead of using them as query statistics to derive the LiRA test statistic as in Equation 2.

We propose an attack that leverages the LiRA framework with an improved query statistic to directly exploit LDAs generative process. Our query statistic is a heuristic for the target document's log-likelihood under LDA. To compute this statistic on $\Phi$ for a given document $d$, the adversary maximizes the log-likelihood of the document over all possible document-topic distributions such that:

$$\zeta(\Phi, d) = \max_{\theta} \ \log p(d|\theta, \Phi)$$
$$= \max_{\theta} \sum_{w \in d} \log(\sum_{z} \theta_z * \Phi_{z,w}), \tag{3}$$

where we optimzie over $\theta \in [0,1]^k$ s.t. $\sum_z \theta_z = 1$. In practice, we use SciPy's out-of-the-box optimization methods to estimate $\zeta(\Phi, d)$ (Virtanen et al., 2020).

Compared to the statistics proposed by Huang et al. (2022), our statistic more directly exploits LDAs generative processes and is based on the observation that a document's likelihood under the target model increases when included in the training data. Therefore, using $\zeta$ allows us to better reason about memorization in topic models. We verify our criteria for the proposed statistic in Appendix C. Additionally, Appendix C contains an evaluation of other candidate statistics where we show that our statistic significantly outperforms other statistics based on the attacks in Huang et al. (2022).

The LiRA with statistic $\zeta$ directly exploits memorization and per-example hardness. The attack accounts for the natural differences in $\zeta$ on various documents by estimating the likelihood ratio under distributions based on $\tilde{\mathbf{T}}_{in}(d)$ and $\tilde{\mathbf{T}}_{out}(d)$. This enables the LiRA to outperform simple attacks like in Huang et al. (2022) that use global thresholds on the other query statistics based on $\hat{\theta}$. Therefore, we propose the LiRA with statistic $\zeta$ as a stronger alternative to attack topic models.

### 3.4 Memorization and Per-Example Hardness

To verify aspects of memorization and per-example hardness for probabilistic topic models, we estimate and visualize $\tilde{\mathbf{T}}_{in}(d)$ and $\tilde{\mathbf{T}}_{out}(d)$ in an experiment similar to Carlini et al. (2022) and Feldman & Zhang (2020). Consistent with their findings, we show that outlying and hard-to-fit observations tend to have a large effect on the learned model when included in the training set for LDA. The histograms in Figure 1 display the estimated distributions for various types of documents.

When included in the training data for $\Phi$, longer and outlying documents increase $\zeta(\Phi, d)$. We note that the word probability $\Phi_{z,w}$ in certain topics dramatically changes for words that appear infrequently in $D$ or words that occur many times in $d$. Together, these factors affect the learned topic model $\Phi$ and shift $\zeta(\Phi, d)$.

The simple fact that the model is more likely to generate specific documents after inclusion in the training data hints toward memorization. Because the generative process for documents in LDA is extremely simple compared to neural language models, they do not learn verbatim sequences of words. Instead, topic model's ability to "memorize" the training data is due to learning based on word co-occurrences.

Long documents are inherently harder to fit than shorter documents. Because the log-likelihood for a document $d$ is the sum of independent log-probabilities for generating each word in $d$, the magnitude of $\zeta(\Phi, d)$ is naturally higher for longer documents. Additionally, the model may struggle to capture coherent topic structures for longer documents because they tend to contain words that span a wider range of topics.

The LiRA turns our observations on memorization and per-example hardness into a MIA. When the words in a document become more common in the training set, $\Phi$ will reflect the new word co-occurrences in a few specific topics which tends to increase $\zeta(\Phi, d)$. This effect is amplified when the document contains many rare words from the vocabulary set. Therefore, the attacker can easily differentiate between $\tilde{\mathbf{T}}_{in}(d)$ and $\tilde{\mathbf{T}}_{out}(d)$ for long and outlying documents.

### 3.5 Attack Evaluation Set-Up

We evaluate our attack against three datasets: *TweetRumors*, *20Newsgroup* and *NIPS*[1]. We apply standard text pre-processing procedures for each dataset: removal of all non-alphabetic characters, tokenization, removal of stop words, removal of tokens longer than 15 characters, removal of tokens shorter than 3 characters, and lemmentization.[2] Appendix D provides the basic data profile for each dataset after pre-processing.

---

[1]The evaluation code and source for each dataset is included in the Availability section (5).

[2]Tokenization, stop words, and lemmatization implemented via python package nltk `https://www.nltk.org`

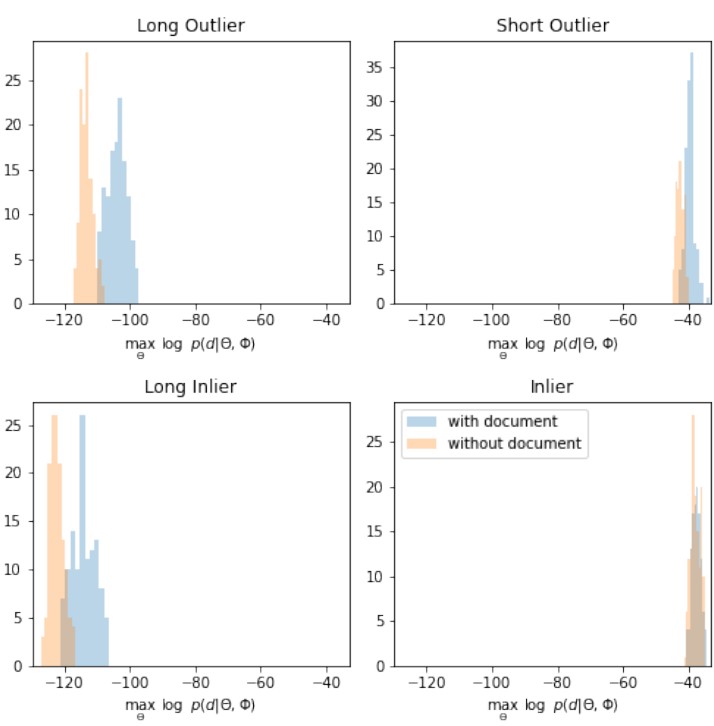

Figure 1: Histograms of the statistic $\zeta(\Phi, d)$ evaluated on different types of documents in *TweetRumors* when $d \in D_{train}$ (blue) and when $d \notin D_{train}$ (orange). Outliers are documents that contains many words that appear infrequently in $D$ and inliers contain many words that appear frequently in $D$. Long documents contain more words than a standard deviation away from the mean document length and short documents contain less. The word count for the inlying document is within one standard deviation of the mean.

Table 1: Attack TPR at 0.1% FPR with 128 Shadow Models

| Dataset | Our Attack Online LiRA | Huang et al. Huang et al. (2022) | | |
|---|---|---|---|---|
| | | Maximum Posterior | Standard Deviation | Entropy |
| *TweetRumors* | 12.8% | 0.19% | 0.19% | 0.18% |
| *NIPS* ($k = 10$) | 44.9% | 1.69% | 1.69% | 1.31% |
| *20Newsgroup* | 21.1% | 0.57% | 0.57% | 0.53% |

To initiate the attack, we randomly sample half of the data to learn and release $\Phi_{obs}$. Next, we train $N$ shadow topic models by repeatedly sampling half of the data to simulate sampling from $\mathbb{D}$. Like in Carlini et al. (2022), this setup creates some overlap between shadow model training data and target model training data which presents a strong assumption made to accommodate the smaller size of our datasets. We do not expect performance to significantly decrease using disjoint datasets as observed by Carlini et al. (2022).

We compare our LiRA against each of the attacks presented by Huang et al. (2022). To replicate their attacks, we randomly sample half of the dataset to learn $\Phi_{obs}$. Using $\Phi_{obs}$, we estimate each documents' topic-distribution $\hat{\theta}$ and compute the maximum posterior, standard deviation, and entropy on $\hat{\theta}$. We directly threshold the each of these statistics to evaluate membership for every document in the dataset.

For each experiment, we learn $\Phi$ using scikit-learn's implementation of LDA with default learning parameters.[3] For *TweetSet* and *20Newsgroup* we set the number of topics $k$ to 5 and 20 respectively based on the data's known distribution across topics: *TweetSet* contains tweets collected on 5 major news events and *20Newsgroup* contains newsgroup documents across 20 different newsgroups. For *NIPS*, we vary $k$ for experimentation purposes. In practice, $k$ is known by the attacker who can observe the dimensions of $\Phi$. We learn $N = 128$ shadow models, replicate each experiment 10 times and report our results across all iterations.

We interpret $\Lambda$ as a predicted membership score where a higher value indicates that the document is more likely to be a member of the training data. We empirically estimate our attacks' performance by evaluating the TPR at all FPRs and plot the attack's ROC curve on log-scaled axes.

### 3.6 Attack Evaluation Results

First, we compare our online attack performance against the attacks in Huang et al. (2022) at an FPR of 0.1% in Table 1. Figure 2 displays the ROC curves for the online and offline variant of our attacks. Table 1 and Figure 2 demonstrate that our LiRA outperforms each of attacks in Huang et al. (2022).

Because the LiRA considers the likelihood-ratio of $d$, it accounts for per-example hardness and dominates the attacks by Huang et al. (2022) at all FPR's. As noted by Carlini et al. (2022), attacks that directly apply global thresholds do not consider document level differences on the learned model and fail to confidently identify members of the training data. Stronger attacks, like our LiRA, should be used to empirically evaluate the privacy associated with topic models.

To understand how the number of topics in $\Phi$ influence attack performance, we vary the number of topics $k$ on *NIPS* and attack the resulting model. Table 2 shows the the attack performance as we vary $k$. As $k$ increases, we see that TPR increases at an FPR of 0.1%. Additionally, we note minor differences between online and offline attack performance.

The number of parameters in $\Phi$ grows linearly by the length of the vocabulary set as $k$ increases. With more topics and more parameters, document's word co-occurrence structure is typically better represented in the document's learned topic distribution. Consequently, increasing $k$ enhances the impact of including the document in the training data resulting in better attack performance.

Overall, our findings demonstrate that despite their simple architecture, topic models exhibit behavior that resembles memorization and are vulnerable to strong MIAs.

---

[3]https://scikit-learn.org/

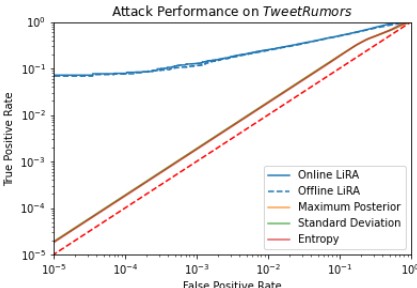 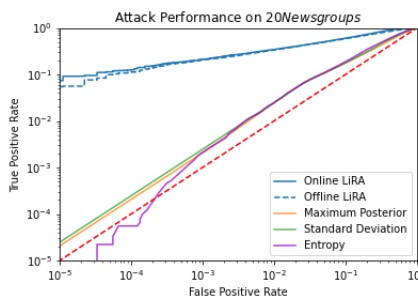 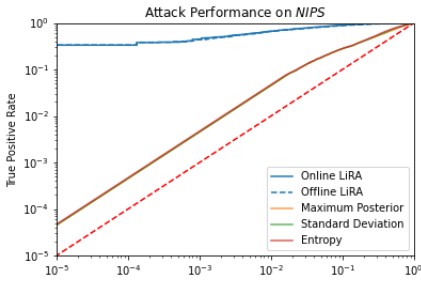

Figure 2: Online and Offline ROC Attack Comparison on Each Dataset (128 Shadow Models, NIPS $k$=10)

Table 2: Attack TPR at FPR of 0.1% While Varying $k$ on *NIPS*

| Number of Topics $k$ | Online LiRA | Offline LiRA |
|---|---|---|
| 5 | 31.8% | 31.8% |
| 10 | 44.9% | 43.6% |
| 15 | 60.5% | 56.0% |
| 20 | 67.9% | 63.2% |
| 25 | 72.5% | 70.2% |

## 4 Private Topic Modeling

Designing private ML solutions is an active field of research. In topic modeling, the literature is largely focused on differential privacy (DP). DP acts as a direct defense against privacy attacks like MIAs by limiting the effect one data point can have on the learned model. Furthermore, DP provides a clear, quantifiable method for reasoning about privacy loss.

### 4.1 Differential Privacy

DP provides strong theoretical guarantees of individual-level privacy by requiring the output of some data analysis to be indistinguishable (with respect to a small multiplicity factor) between any two adjacent datasets $x$ and $x'$.

**Definition 4.1 (Differential Privacy (Dwork et al., 2006b;a))** Let $M : \mathcal{X}^n \to \mathcal{R}$ be a randomized algorithm. For any $\varepsilon \geq 0$ and $\delta \in [0,1]$, we say that $M$ is $(\varepsilon, \delta)$-differentially private if for all adjacent databases $x, x' \in \mathcal{X}^n$ and every $S \subseteq \mathcal{R}$

$$\Pr[M(x) \in S] \leq e^\varepsilon \Pr[M(x') \in S] + \delta.$$

The notion of adjacency is critical. For any text dataset, we say that two corpora $D$ and $D'$ are *author-level adjacent* if they differ by one author's documents. This notion enforces DPs promise of individual level privacy by considering adjacency at the user level.

If two corpora $D$ and $D'$ differ by one document, then adjacency is at the *document-level*. This relaxed notion of adjacency can satisfy author-level adjacency if we assume that each document has a unique author. For corpora the most relaxed notion of adjacency is *word-level adjacency*. Two corpora $D$ and $D'$ are word-level adjacent if they differ by one word in one document.

To achieve differential privacy we must add noise whose magnitude is scaled to the worst-case sensitivity of the target statistics from all adjacent datasets. Researchers have explored word-level adjacency to control sensitivity, but this approach offers weak privacy guarantees. For instance, most DP collapsed Gibbs sampling algorithms for learning LDA rely on perturbing word-count statistics by adding noise scaled to word-level adjacency (Zhao et al., 2021; Huang & Chen, 2021; Huang et al., 2022).

There are a variety of other algorithms for DP topic modeling: Park et al. (2016) propose DP stochastic variational inference to learn LDA, Decarolis et al. (2020) use a privatized spectral algorithm to learn LDA, and Wang et al. (2022) satisfy DP by adding noise directly to $\Phi$. These algorithms satisfy various notions of DP and use differing notions of adjacency. Table 5 in Appendix E summarizes each DP topic modeling implementation.

While each DP topic modeling algorithm has their advantages and disadvantages, there are some common themes across implementations. First, the iterative nature of learning algorithms for LDA posses a difficult composition issue when managing the privacy loss. Next, many implementations use relaxed notions of adjacency to control sensitivity. Finally, none of the existing methods address the privacy concerns with releasing the vocabulary set corresponding to $\Phi$.

## 4.2 DP Vocabulary Selection

Releasing the topic-word distribution $\Phi$ without ensuring the privacy of the accompanying vocabulary set can lead to privacy violations because the vocabulary set is derived directly from the data. Topic models with comprehensive vocabulary sets tend to include many infrequently used words from the corpus which makes them more vulnerable to MIAs (as shown in section 3.4). Furthermore, even if the values in $\Phi$ are private, the overall release of the topic-word distribution can not satisfy DP because the vocabulary set accompanied by $\Phi$ is not private.

To illustrate this point, consider a scenario where we have a corpus $D$ with a single document $d$. If we apply DP topic modeling to release $\Phi$ without changing the vocabulary set, then we release all of the words in $d$. This clearly compromises the privacy of the document in $D$. If we chose not to release the vocabulary set, $\Phi$ loses practical interpretability. To address this issue, it is necessary to explore methods for differentially private vocabulary selection.

DP vocabulary selection can be formalized as the DP Set-Union (DPSU) (Gopi et al., 2020). If we assume each author $i$ contributes a subset $W_i \subseteq U$ of terms in their documents, DPSU seeks to design a $(\varepsilon, \delta)$-DP algorithm to release a vocabulary set $S \subseteq \cup_i W_i$ such that the size of $S$ is as large as possible. Researchers first approached this problem with the intent to release the approximate counts of as many items as possible in $\cup_i W_i$ (Korolova et al., 2009; Wilson et al., 2020). Their algorithms guarantee privacy by constructing a histogram of $\cup_i W_i$, adding noise to each word frequency in the histogram, and releasing all word frequencies that fall above a certain threshold. These implementations upper-bound the sensitivity based on the maximum the number of words contributed by all users ($\Delta_0 = \max_i |W_i|$) to account for users contributing more than one word. However, most users contribute significantly less than $\Delta_0$ so this method wastes sensitivity. Gopi et al. (2020) propose algorithms that build weighted histograms with update policies that help control sensitivity. Intuitively, the update policy stops increasing the weights of words in the histogram after they reach some cutoff $\Gamma$ which is set using the parameter $\alpha$. Carvalho et al. (2022) propose an improved algorithm for DPSU which is well fit for vocabulary selection because it allows for one user to contribute the same term multiple times.

## 4.3 Fully Differentially Private Topic Modeling

We propose a high-level procedure for fully DP topic modeling (FDPTM), in Algorithm 1. FDPTM composes the privately selected vocabulary set $S$ and the private topic-word distribution $\Phi$. Theorem 4.1 states the privacy guarantees associated with FDPTM. The proof is deferred to Appendix F.

---
**Algorithm 1** Fully Differentially Private Topic Modeling (FDPTM)

---
1: **Require:** Corpus $D$, DP vocabulary selection algorithm $M_1$ and DP topic modeling algorithm $M_2$.
2: Apply standard pre-processing and tokenization: $D_{pre} \leftarrow \texttt{PRE}(D)$
3: Apply DP vocabulary selection to corpus: $S \leftarrow M_1(D_{pre})$
4: Remove all words $w \in D_{pre}$ if $w \notin S$: $D_{san} \leftarrow \texttt{SAN}(D_{pre}, S)$
5: Learn $\Phi$ using DP topic modeling: $\Phi \leftarrow M_2(D_{san})$
6: **Release:** Topic-word distribution $\Phi$ and corresponding vocabulary set $S$

---

**Theorem 4.1 (Privacy Guarantee of FDPTM)** If $M_1$ for selecting the vocabulary set satisfies $(\varepsilon_1, \delta_1)$-DP and $M_2$ for topic modeling satisfies $(\varepsilon_2, \delta_2)$-DP, then the overall release of $\Phi$ satisfies $(\varepsilon_1 + \varepsilon_2, \delta_1 + \delta_2)$-DP.

Implementing FDPTM requires a few key considerations. First, the process depends on carefully tuning the privacy parameters and other parameters within $M_1$ and $M_2$. Second, the data curator must ensure that $M_1$ and $M_2$ satisfy the same notion of differential privacy and adjacency.

Overall, FDPTM is a modular framework for releasing meaningful topic-word distributions. Although our evaluations focus on LDA, other topic models for specific uses can easily fit into the proposed procedure.

### 4.4 FDPTM Evaluation Set-Up

We empirically evaluate our FDPTM using the *TweetRumors* dataset with $k = 5$. We ensure author-level privacy via document-level adjacency under the assumption that each document has a unique author. This assumption necessary for controlling sensitivity and is common to most DP NLP solutions. We accomplish this by using the DPSU solution proposed by Carvalho et al. (2022) for vocabulary selection and DP LDA learning algorithm by Zhu et al. (2016) because they satisfy the same notion of privacy. For DPSU, we fix the privacy parameter $\delta = 10^{-5}$ and set the parameter $\alpha = 3$ to control the cut-off value $\Gamma$ like in Carvalho et al. (2022).

To evaluate utility, we first vary the privacy loss parameter for DPSU and analyze utility when LDA is non-private. We study the interaction between DP vocabulary selection and DP LDA by fixing the privacy loss parameter for one mechanism and varying the privacy loss parameter for the other mechanism. Specifically, we first fix $\varepsilon_2 = 3$ for DP LDA and vary $\varepsilon_1$ for DPSU. Then, we fix $\varepsilon_1 = 3$ and vary $\varepsilon_2$. We vary our privacy loss parameters in intervals across common $\varepsilon$ choices for DP ML solutions. Typically, $\varepsilon > 1$ provides very weak theoretical privacy guarantees, but tend to be effective for providing empirical privacy.

We evaluate the utility of the FDPTM using topic coherence (Mimno et al., 2011), which can be a useful proxy for measuring the interpretability of topics. Topic coherence for a topic $t$ is defined as

$$coherence(t; V^{(t)}) = \sum_{m=2}^{M} \sum_{l=1}^{m-1} \log \frac{D(v_m^{(t)}, v_l^{(t)}) + 1}{D(v_l^{(t)})}, \tag{4}$$

where $D(v)$ is the document frequency of word $v$ (i.e. the number of documents where word $v$ occurs), $D(v, v')$ is the co-document frequency of words $v$ and $v'$ (i.e. the number of documents where $v$ and $v'$ both occur), and $V^{(t)} = \{v_1^{(t)}, ..., v_M^{(t)}\}$ is list of the $M$ most probable words in topic $t$. A high topic coherence score suggests that the top $M$ terms have higher co-document frequencies which makes the topic easier to understand because the terms are more semantically similar or coherent. In our analysis, we select each topic's top $M = 10$ words and report the average across each topic.

We also test FDPTM against the online LiRA to empirically evaluate the privacy of FDPTM. For each attack we learn 64 shadow models. Like before, we conduct attacks while varying $\varepsilon_1 \in \{1, 5, 10\}$ and fixing $\varepsilon_2 = 5$. Then, we fix $\varepsilon_1 = 5$ and vary $\varepsilon_2$. We repeat each experiment 10 times and report the results across each iteration.

### 4.5 FDPTM Evaluation Results

Figure 3 displays topic coherence as we increase $\varepsilon_1$. Figure 4 displays topic coherence as we vary either $\varepsilon$ and hold the other constant. We include the average vocabulary size as we increase $\varepsilon_1$ in Appendix G.

Topic coherence remains stable after $\varepsilon = 3$, suggesting that increasing the privacy loss for DPSU may not affect topic coherence after a certain point. When comparing coherence while increasing $\varepsilon_2$, we see that $\varepsilon_1$ decreases coherence regardless of $\varepsilon_2$. In Figure 4, coherence plateaus around $\varepsilon_1 = 3$ while increasing $\varepsilon_2$ continues to boost coherence. Consequently, increasing $\varepsilon_1$ yields diminishing utility returns faster.

Figure 5 presents the ROC curves that explore the how the interaction between DP vocabulary selection and DP LDA affect attack performance. We gain some empirical privacy under FDPTM by decreasing LiRA

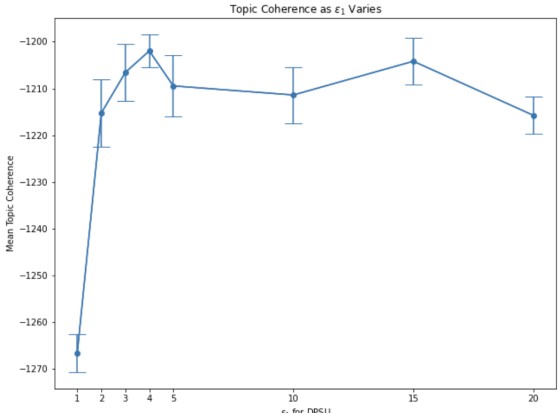

Figure 3: Topic coherence as $\varepsilon_1$ increases and LDA is not private. The non-private baseline for topic coherence is very small compared to the figure ($\approx -1117$). The error bars represent one standard deviation from the mean topic coherence.

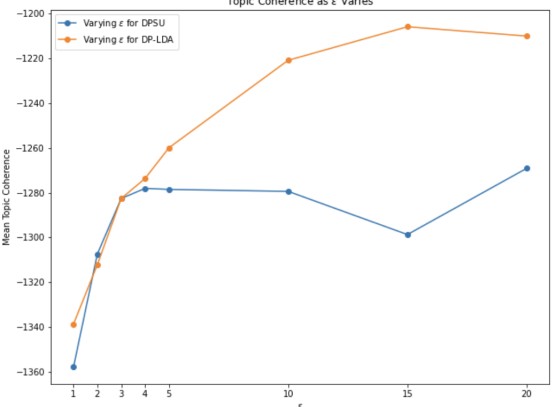

Figure 4: Topic Coherence as $\varepsilon$ Increases. The blue line shows the the results while varying $\varepsilon_1$ for DPSU and holding $\varepsilon_2 = 3$. Orange displays results for varying $\varepsilon_2$ for DP LDA while holding $\varepsilon_1 = 3$.

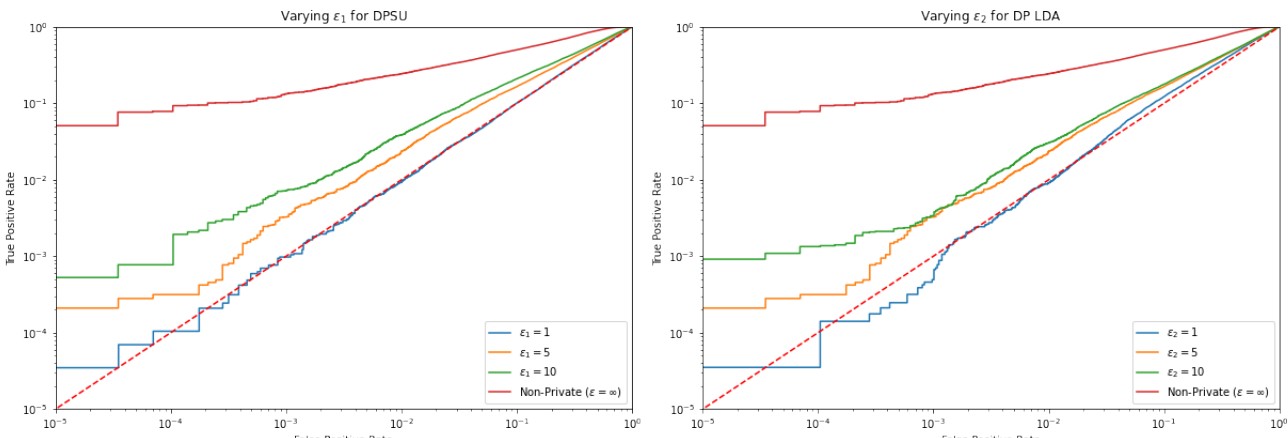

Figure 5: Attack ROCs while varying privacy loss parameters for DPSU and DP LDA. On the right-hand side, we fix $\varepsilon_2 = 5$ for DP LDA and vary $\varepsilon_1 \in \{1, 5, 10\}$ for DPSU. On the left-hand side, we fix $\varepsilon_1$ and vary $\varepsilon_2$ at the same intervals. We provide the non-private baseline in red for reference.

performance at all FPRs. Generally, we see that attack performance decreases similarly as we decrease either privacy parameter. Furthermore, we gain empirical privacy by decreasing LiRA performance at all FPRs.

DPSU decreases attack performance because we shrink the size of the vocabulary set, limiting the number of parameters in $\Phi$, and forcing each documents' topic distribution to look more like other documents' topic distributions. The nature of learning on word co-occurrences forces $\Phi$ to reflect identifiable changes when infrequently used words suddenly become more common to the training data. By removing these words with DPSU, we limit the ability for outliers to have large effects on the topic model.

Our results indicate that dedicating most of the privacy budget to DP LDA, rather than DPSU, increases utility at the same privacy level. Intuitively, we can inject less noise into DP LDA, and more into the vocabulary selection algorithm ($\varepsilon_1 < \varepsilon_2$) to increase model interpretability at the same global privacy loss ($\varepsilon_1 + \varepsilon_2$) and similar empirical privacy against the LiRA. This approach allows us balance privacy and utility in topic modeling.

## 5 Conclusions

In our work, we show that topic models exhibit aspects of memorization and successfully implement strong MIAs against them. Although probabilistic topic models do not memorize verbatim sequences of text like language models, they do memorize word frequency and occurrence. In some instances, knowledge of the frequency of certain terms in a document constitutes a privacy violation regardless of their word ordering.

To combat MIAs and memorization, we propose a modular framework for implementing better private topic models using differential privacy. Overall, the addition of DP vocabulary selection to the DP topic modeling workflow is important for guaranteeing private, interpretable releases of $\Phi$. Not only does DP vocabulary selection allow for fully DP releases of $\Phi$, it also provides an effective defense against MIAs.

We highlight the greater need for continued development in the field of privacy-preserving ML. If simple probabilistic topics models for text memorize their training data, then it's only inevitable that LLMs and neural topic models memorize their training data. As ML models continue to become more sophisticated and widely used, privacy concerns become increasingly relevant.

## Limitations

While our results provide insights on topic models' privacy, there are a few limitations of our methodology and experiments. First, because we rely on a BoW document representation, we only identify that documents with certain word frequencies are part of the training set. In practice, this may cause the attack to realize more false positives. The datasets used in our attack evaluations did not include documents with overlapping word combinations, and most reasonable documents contain very few sensible alternative word combinations which eases the implications of this limitation.

Additionally, our utility assessments for FDPTM are constrained by the limitations of automated topic evaluation methods. While different measures give us different insights into the topic model, the best evaluations are typically based on human inspection (Hoyle et al., 2021). Other utility metrics that evaluate topic models with metrics that estimate model fit, like perplexity, tend to be negatively correlated with human-measured topic interpretability (Chang et al., 2009). Therefore, we use topic coherence because it is associated with the human interpretability of topics.

Finally, while our results are convincing, we conduct them with the same learning process ($f_{\mathcal{G}}$ based on LDA via Gibbs-sampling) and hyperparameters for the the shadow and target models with access to the entire topic-word distribution $\Phi$. In many practical use cases, researchers may not use LDA or release all of $\Phi$. A mismatch between shadow and topic models $f_{\mathcal{G}}$ may impact attack accuracy, but we suspect to a lesser extent than in neural models. Additionally, we believe that clever adversaries can reconstruct $\Phi$ given partial or alternative releases of $\Phi$. Determining the affect that different $f_{\mathcal{G}}$ and releases of $\Phi$ have on attack performance is left for future research.

## Ethics Statement

The privacy attacks presented in this paper are for research purposes only. The methodology and code should be used responsibly with respect for privacy considerations. Each of the datasets used in our experiments are open access and do not pose a direct threat to the privacy of the users who contributed to them.

## Acknowledgement

N.M. is supported through the MIT Lincoln Laboratory (MITLL) Military Fellowship and by NSF grant BCS-2218803. W.Z. is supported by a Computing Innovation Fellowship from the Computing Research Association (CRA) and the Computing Community Consortium (CCC). S.V. is supported by NSF grant BCS-2218803 and a Simons Investigator Award. The authors would like to acknowledge the MITLL Supercomputing Center for providing compute and consulting resources that contributed to our results.

## Availability

The URLs for the datasets *TweetRumors*, *20Newsgroup* and *NIPS* are `https://www.zubiaga.org/datasets`, `http://qwone.com/~jason/20Newsgroups` and `https://archive.ics.uci.edu/ml/datasets/bag+of+words` respectively. The code for the attack simulations is available at `https://github.com/nicomanzonelli/topic_model_attacks`.

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

## A   The Online LiRA

The LiRA on topic models follows directly from the LiRA on supervised ML models proposed by Carlini et al. (2022). However, the attack on topic models differs because we use statistic $\zeta$ queried on $\Phi$. Algorithm 2 contains the online LiRA.

## B   The Offline LiRA

The offline LiRA on topic models also follows directly from the offline LiRA on supervised ML models proposed by Carlini et al. (2022). The offline variant benefits over the online variant because lines 3-9 can

---

**Algorithm 2** Online LiRA on Topic Models

---
1: **Inputs:** Target topic-word distribution: $\Phi_{obs}$, target document: $d$, shadow model iterations: $N$, statistic function: $\zeta$, data distribution: $\mathbb{D}$, and topic model $f_{\mathcal{G}}$.
2: **Returns** Likelihood Ratio Test Statistic: $\Lambda$
3: $\tilde{\mathbf{T}}_{in} = \{\}$                  $\triangleright$ Initialize empty sets
4: $\tilde{\mathbf{T}}_{out} = \{\}$
5: **for** $N$ times **do**
6:    $D \leftarrow \mathbb{D}$             $\triangleright$ Sample auxiliary data from distribution
7:    $\Phi_{in} \leftarrow f_{\mathcal{G}}(D \cup \{d\})$         $\triangleright$ Train shadow model with target document
8:    $\tilde{\mathbf{T}}_{in} \leftarrow \tilde{\mathbf{T}}_{in} \cup \{\zeta(\Phi_{in}, d)\}$
9:    $\Phi_{out} \leftarrow f_{\mathcal{G}}(D \setminus \{d\})$       $\triangleright$ Train shadow model without target document
10:    $\tilde{\mathbf{T}}_{out} \leftarrow \tilde{\mathbf{T}}_{out} \cup \{\zeta(\Phi_{out}, d)\}$
11: **end for**
12: $\mathcal{N}_{in} \leftarrow \mathcal{N}(\text{mean}(\tilde{\mathbf{T}}_{in}), \text{var}(\tilde{\mathbf{T}}_{in}))$       $\triangleright$ Estimate normals
13: $\mathcal{N}_{out} \leftarrow \mathcal{N}(\text{mean}(\tilde{\mathbf{T}}_{out}), \text{var}(\tilde{\mathbf{T}}_{out}))$
14: $\Lambda \leftarrow \frac{p(\zeta(\Phi_{obs}, d) \mid \mathcal{N}_{in})}{p(\zeta(\Phi_{obs}, d) \mid \mathcal{N}_{out})}$         $\triangleright$ Calculate test statistic
15: **Return** $\Lambda$

---

be executed regardless of the target document $d$, and lines 11-17 can be repeated for any $d$ after estimating $\mathbf{T}_{out}$ once. Additionally, note that the test statistic $\Lambda$ changes in line 15. Algorithm 3 contains the offline LiRA.

---

**Algorithm 3** Offline LiRA on Topic Models

---
1: **Inputs:** Target topic-word distribution: $\Phi_{obs}$, target document: $d$, shadow model iterations: $N$, statistic function: $\zeta$, data distribution: $\mathbb{D}$, and topic model $f_{\mathcal{G}}$.
2: **Returns** Likelihood Ratio Test Statistic: $\Lambda$
3: $\mathbf{T}_{out} = \{\}$                  $\triangleright$ Initialize empty set
4: **for** $N$ times **do**
5:    $D \leftarrow \mathbb{D}$             $\triangleright$ Sample auxiliary data from distribution
6:    $\Phi_{out} \leftarrow f_{\mathcal{G}}(D)$          $\triangleright$ Train shadow models without target document
7:    $\mathbf{T}_{out} \leftarrow \mathbf{T}_{out} \cup \{\Phi_{out}\}$
8: **end for**
9: $\tilde{\mathbf{T}}_{out} = \{\}$              $\triangleright$ Calculate Statistic Across $\mathbf{T}_{out}$
10: **for** $\Phi_{out}$ in $\mathbf{T}_{out} = \{\}$ **do**
11:    $\tilde{\mathbf{T}}_{out} \leftarrow \tilde{\mathbf{T}}_{out} \cup \{\zeta(\Phi_{out}, d)\}$
12: **end for**
13: $\mathcal{N}_{out} \leftarrow \mathcal{N}(\text{mean}(\tilde{\mathbf{T}}_{out}), \text{var}(\tilde{\mathbf{T}}_{out}))$        $\triangleright$ Calculate Test Statistic
14: $\Lambda \leftarrow 1 - \text{Pr}[\mathcal{N}_{out} > \zeta(\Phi_{obs}, d)]$
15: **Return** $\Lambda$

---

## C  Evaluating Candidate Query Statistics

In this section, we provide analysis for the proposed statistic $\zeta$ w.r.t. the two query statistic requirements detailed in 3.3:

1. the statistic should increase for $d$ when $d$ is included in the training data ($\tilde{\mathbf{T}}_{in}(d) > \tilde{\mathbf{T}}_{out}(d)$) to enable the offline LiRA.

2. The estimated distributions $\tilde{\mathbf{T}}_{in}(d)$ and $\tilde{\mathbf{T}}_{out}(d)$ are approximately normal to allow for parametric modeling.

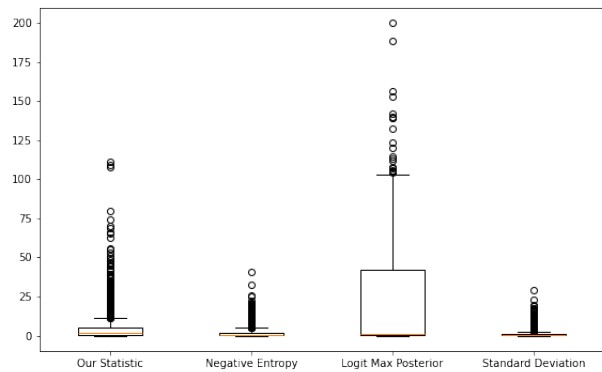

Figure 6: Boxplot of KL Divergences between $\mathcal{N}_{in}(d)$ and $\mathcal{N}_{in}(d)$ evaluated on 3000 documents.

Table 3: Shapiro-Wilk Tests with FDR Control for 6000 Tests on Each Statistic ($\alpha = .05$)

| Statistic | Number of Rejections | Rejection % |
|---|---|---|
| Our Statistic | 1335 | 22.25% |
| Negative Entropy | 3584 | 59.73% |
| Logit Max Posterior | 5191 | 86.52% |
| Standard Deviation | 4621 | 77.02% |

Additionally, we compare the proposed statistic to alternative statistics based on the attacks in Huang et al. (2022) and show that our proposed statistic outperforms the alternatives.

Huang et al. (2022) propose three statistics: the entropy of $\hat{\theta}$, the standard deviation of $\hat{\theta}$, and the maximum value in $\hat{\theta}$ where $\hat{\theta}$ is a document's estimated topic distribution Huang et al. (2022). Each statistic requires estimating the document's topic distribution $\hat{\theta}$. They estimate $\hat{\theta}$ using sampling techniques, but for these experiments, we choose to estimate $\hat{\theta}$ by performing the optimization from Eq. 3 for increased efficiency and consistency with the proposed statistic in Eq. 3.

To tailor each statistic to fit within the LiRA framework, we make some minor modifications to the statistics proposed by Huang et al. (2022). Because entropy naturally decreases when the document is included in the training data, we use the negative entropy to satisfy the first requirement. Additionally, because the maximum value in $\hat{\theta}$ is bound between [0,1] and tends to concentrate toward 1 for outliers, we apply the logit transformation to promote adherence to the normality requirement, as illustrated by Carlini et al. (2022).

To evaluate the query statistic selection criteria, we fit an online attack using $N = 256$ shadow models using each of our candidate statistics. We sample 1000 documents from each dataset to extract $\tilde{\mathbf{T}}_{in}(d)$ and $\tilde{\mathbf{T}}_{out}(d)$ for each candidate statistic. We assess the first requirement by evaluating the KL-divergence between two normal distributions estimated from $\tilde{\mathbf{T}}_{in}(d)$ and $\tilde{\mathbf{T}}_{out}(d)$. A positive KL-divergence between $\mathcal{N}_{in}(d) = \mathcal{N}(\text{mean}(\tilde{\mathbf{T}}_{in}(d)), \text{var}(\tilde{\mathbf{T}}_{in}(d)))$ and $\mathcal{N}_{out}(d) = \mathcal{N}(\text{mean}(\tilde{\mathbf{T}}_{out}(d)), \text{var}(\tilde{\mathbf{T}}_{out}(d)))$ indicates that the statistic is greater when $d$ is included in the training data. Figure 6 displays a boxplot for the KL divergences for each statistic. We note that all KL-divergences in Figure 6 are greater than 0 which indicates that $\tilde{\mathbf{T}}_{in}(d) > \tilde{\mathbf{T}}_{out}(d)$.

We evaluate the normality requirement by performing a Shipiro-Wilk test for each $\tilde{\mathbf{T}}_{in}(d)$ and $\tilde{\mathbf{T}}_{out}(d)$ with control for the False Discovery Rate (FDR) using the Benjamini-Hochberg Procedure (Shapiro & Wilk, 1965; Benjamini & Hochberg, 1995). The null hypothesis is that the data is drawn from a normal distribution. Table 3 contains the the number of tests rejected with FDR control for each statistic at a significance level of .05. Our analysis indicates that the distributions of $\tilde{\mathbf{T}}_{in/out}(d)$ for our statistic tend to be normal more often than the alternatives. We can expect to encounter fewer rejections as we increase $N$ (i.e. increase the sample size of $\tilde{\mathbf{T}}_{in/out}(d)$).

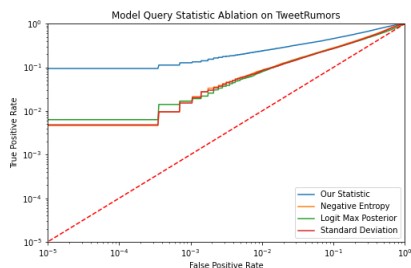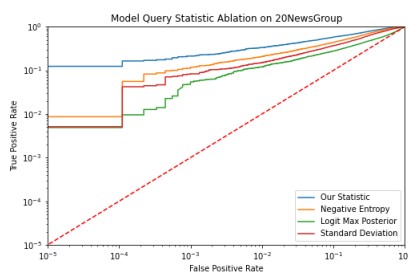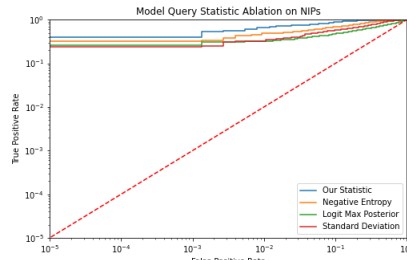

Figure 7: Online Attack Comparison Across Statistics on Each Dataset (256 Shadow Models, NIPS $k$=10)

Table 4: Dataset Profile After Pre-Processing

| Dataset | Number of Documents $M$ | Average Document Length | Vocabulary Size $V$ |
|---|---|---|---|
| *TweetRumors* | 5,698 | 9 | 5,942 |
| *NIPS* | 1,494 | 893 | 10,346 |
| *20Newsgroup* | 18,037 | 84 | 74,781 |

Table 5: A Brief Summary of the Existing Literature on DP Topic Modeling

| Authors | Notion of DP | Notion of Adjacency | Learning Method | Other Technical Details |
|---|---|---|---|---|
| Zhu et al. (2016) Zhu et al. (2016) | $\varepsilon$-DP | document-level | CGS | |
| Zhao et al. (2021) Zhao et al. (2021) | $\varepsilon$-DP | word-level | CGS | |
| Huang and Chen (2021) Huang & Chen (2021) | $(\varepsilon, \delta)$-DP | word-level | CGS | sub-sampling |
| Huang et al. (2022) Huang et al. (2022) | Rényi-DP | word-level | CGS | |
| Park et al. (2018) Park et al. (2016) | $(\varepsilon, \delta)$-DP | document-level | VI | moments accountant and sub-sampling |
| Decarolis et al. (2020) Decarolis et al. (2020) | Rényi-DP | document-level | SA | local sensitivity from propose-test-release |
| Wang et al. (2022) Wang et al. (2022) | Rényi-DP | author-level | PH | pain-free smooth sensitivity |

Finally, in Figure 7, we evaluate the attack performance of all statistics across each dataset. We show that the LiRA with our proposed statistic dominates the other candidate statistics at all FPRs. While each of the candidate statistics satisfies requirement 1, the long-whiskers and outliers associated with the logit maximum posterior statistic in Figure 6 suggest that it would outperform other statistics because the model can more easily differentiate between $\mathcal{N}_{in}(d)$ and $\mathcal{N}_{out}(d)$. However, the statistics' lack of normality seems to hinder performance.

## D  Data Profile

Table 4 contains the data profile for each dataset after pre-processing.

## E  Existing DP Topic Models

Table 5 summarizes centralized DP topic modeling algorithms available in the literature. The learning methods included are Collapesed Gibbs Sampling (CGS), variational inference (VI), spectral algorithm (SA), and Model Agnostic or Post-Hoc (PH).

## F  Proof of Theorem 4.1

First, let $M_1$ be a $(\varepsilon_1, \delta_1)$-DP vocabulary selection algorithm that returns a private vocabulary set $S$, and $M_2$ be a $(\varepsilon_2, \delta_2)$-DP topic modeling algorithm that returns a private topic-word distribution $\Phi$. Now, let `PRE` be a function that takes a corpus $D$ and applies standard pre-processing procedures, and `SAN` be a function

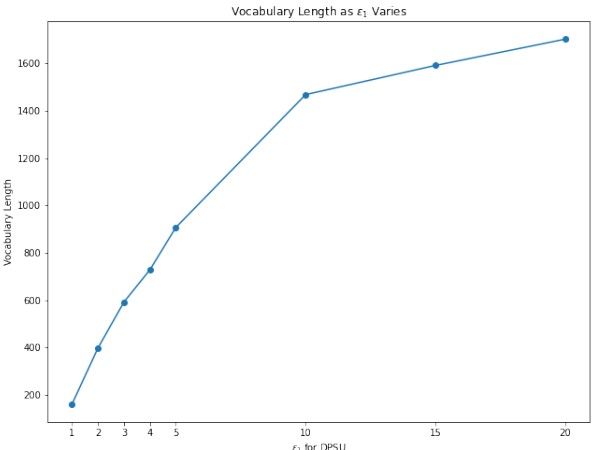

Figure 8: The length of the vocabulary set as $\varepsilon_1$ for DPSU increases. The original vocabulary size is 5,942.

that takes a corpus $D$ and a vocabulary set $S$ and removes all words $w \in D$ if $w \notin S$. Finally, let $M_x$ represent the FDPTM algorithm such that for a corpus $D$

$$M_x = (M_1(\texttt{PRE}(D)),$$
$$M_2(\texttt{SAN}(\texttt{PRE}(D),\ M_1(\texttt{PRE}(D)))))).$$

**Definition F.1 ($k$-Stability (Thakurta & Smith, 2013))** A function $f : U^* \to \mathcal{R}$ is $k$-stable on input $D$ if adding or removing any $k$ elements from $D$ does not change the value of $f$, that is, $f(D) = f(D')$ for all $D'$ such that $D \triangle D' \leq k$. We say $f$ is stable on $D$ if it is (at least) 1-stable on D, and unstable otherwise.

**Lemma F.1 (Composition with Stable Functions (Thakurta & Smith, 2013))** Let $f$ be a stable function, and let $M$ be an $(\varepsilon, \delta)$-DP algorithm. Then, their composition $M(f(x))$ satisfies $(\varepsilon, \delta)$-DP.

The functions `PRE` and `SAN` are stable algorithms because each document is processed independently of the others based on a standard set of rules. Simply, adding or removing a document from the corpus does not affect the functions behavior on other documents,. Therefore, `PRE` and `SAN` are stable functions.

Via our definition of $M_1$, and using the fact that `PRE` is a stable function, then the first term in $M_x$, $M_1(\texttt{PRE}(D))$, satisfies $(\varepsilon_1, \delta_1)$-DP. The second term of $M_x$ applies $M_2$ which directly depends on $M_1$ and the stable functions `PRE` and `SAN`. Therefore, via adaptive composition, $M_x$ is $(\varepsilon_1 + \varepsilon_2, \delta_1 + \delta_2)$-DP (Dwork et al., 2006a). $\square$

# G Vocabulary Size as $\varepsilon_1$ Increases

Figure 8 contains a plot for the size of the vocabulary set as the privacy loss parameter $\varepsilon_1$ for DPSU increases.

