# OpenReview forum: "Membership Inference Attacks and Privacy in Topic Modeling"
_TMLR — Accepted by TMLR_

### Review · Reviewer_hNKb · 2024-05-03

**Summary Of Contributions:**

This paper studies privacy in topic models, specifically the Latent Dirichlet Allocation (LDA). Firstly, a membership inference attack (MIA) based on on the Likelihood Ratio Attack (LiRA) framework is proposed against topic models to show that an adversary can infer membership of documents in the training data. The experiments show that using the proposed attack, an adversary can predict the membership of a certain document in the training data  especially as the number of topics increases. Moreover, the experiments show that the proposed attack outperforms the previously used MIA attacks proposed by Huang et. al. (2022).

As a second step, a framework is proposed to incorporate differential privacy (DP) to the topic modeling algorithm where DP is used in the vocabulary selection, as well as in the learning of the model.

**Audience:**

Yes

**Broader Impact Concerns:**

The broader impacts are sufficiently addressed.

**Claims And Evidence:**

Yes

**Requested Changes:**

The changes requested are directly related to the weaknesses stated in the previous section:
-  Please add more description of the methods from literature that are used in this work so the paper is self contained and more understandable.
- Please explain how the values are picked in the experiments.
- Please fix the typos/errors stated above.

It is important that the above comments are addressed, or if the authors disagree, that it is explained why it is not possible or not necessary in their view.

**Strengths And Weaknesses:**

Understanding the privacy concerns of topic modeling is an interesting and important topic. The paper is generally well written and has good experiments showing the effectiveness of the proposed frameworks.

However, the paper would benefit from having more description of the methods from literature that are used in this work, for instance the DPSU solution and the DP LDA algorithm used in section 4.2 and 4.4.

Moreover, the values used for the experiments seem random, it is unclear why those values are chosen, for instance the number of topics in section 3.5 and the values of $\epsilon_1$ and $\epsilon_2$ in section 4.4.

Some additional questions:
- In table 2, it is said that "we see that TPR increases at all FPRs", but the FPR is 0.1%, so how is this at all FPRs?
- How restrictive is assuming that each author has a single document and are there any ideas on how expensive/messy it would be to break this assumption?

Additionally, there are some minor comments:
- In the introduction:
     * "... the membership of documents included the training data..." -> "... the membership of documents included _in_ the training data..."
     * "We propose an algorithm for DP topic modeling provides privacy..." -> "We propose an algorithm for DP topic modeling _that_ provides privacy..."
- In section 2.3: "... fail to show that their attack confidently identify members of the training data." -> "... fail to show that their attack confidently identifies members of the training data."
- Section 3.2: "Hence, $\zeta$ must be careful tailored to topic models for the attack to be effective" -> "Hence, $\zeta$ must be carefully tailored to topic models for the attack to be effective"
- In section 3.4:
     * "The simple fact that model is more likely ..." -> "The simple fact that _the_ model is more likely ..."
     * "Long document's" -> "Long documents"
- Section 3.5: Lemmentization, do you mean Lemmatization?

---

> ### Author Response · Authors · 2024-07-07
>
> Thank you for your comments. We provide an answer to a few of your questions below.
>
> - **In table 2, it is said that "we see that TPR increases at all FPRs", but the FPR is 0.1\%, so how is this at all FPRs?** Thank you for pointing out this discrepancy. This statement is an artifact from a previous draft where we included a crowded ROC curve to visualize the results while varying $k$. The revised copy simplifies ROC curve to a table and the statement to "As $k$ increases, we see that TPR increases at an FPR of 0.1\%"
>
> - **How restrictive is assuming that each author has a single document and are there any ideas on how expensive/messy it would be to break this assumption?** Typically DP NLP/ML literature assumes that document level privacy is sufficient. How realistic or restrictive this assumption is depends largely on the underlying data domain. For instance, assuming that each observation a set of tweets comes from a different user is more realistic than assuming that each observation in a set of company emails comes from a different user. Breaking this assumption is very expensive in-terms sensitivity because the notion of adjacency relies on removing/replacing all of one user's data in the dataset. If the user is allowed an unknown number of documents, then sensitivity is unbounded. We added a brief note in the revisions on this in section 4.4.
>
> To address the reviewers primary concerns we made the following revisions. They appear in blue in the revised copy.
>
> - We describe methods for DPSU or vocabulary solution from the literature in section 4.2. This makes the paper more self-contained and allows the audience to better understand the meaning of $\alpha$ in section 4.4. Additionally, we include a brief description of the LDA algorithm we use for experiments in 4.4 and 4.5. We chose not to add lengthy descriptions of all DP-LDA solutions because it does not add much context for our experiments and it is summarized by the table in the Appendix E.
>
> - We describe how we chose $k$ for experimentation in section 3.5. The description of DPSU solutions also helps assist with interpreting the parameters described section 4.4. Additionally, in section 4.4 we briefly discuss selection of privacy loss parameters $\epsilon$. We choose $\varepsilon$ in experimentation across common $\varepsilon$ choices for DP ML solutions. These choices are typically higher for DP ML solutions because the process is very sensitive to noise. Choosing $\varepsilon > 1$ provides very weak theoretical privacy guarantees, but tend to be effective for providing empirical privacy even up to higher $\varepsilon$ like 10.
>
> - Corrected the typos highlighted by your review and revised the paper for general clarity (not in blue text).

---

> > ### Comment · Reviewer_hNKb · 2024-07-15
> >
> > Thank you for the response and revision.

---

### Review · Reviewer_NtZa · 2024-06-10

**Summary Of Contributions:**

The paper develops a novel summary statistic for membership inference attacks (MIA) on latent Dirichlet allocation (LDA), which empirically outperforms statistics from previous work. The paper also considers differential privacy for LDA. Unlike previous work, the paper considers the vocabulary of the input documents as private information, and uses a DP set union algorithm to release the vocabulary privately. This is combined with a DP LDA algorithm from previous work. The combination of the DP algorithms empirically defends against the MIA, and retains some utility at high values of $\epsilon$.

**Audience:**

Yes

**Broader Impact Concerns:**

The paper includes an ethics statement which covers the relevant issues.

**Claims And Evidence:**

Yes

**Requested Changes:**

Minor changes
- The experimental results should have uncertainty estimates, at least for the tables.
- The performance for the full mechanism would be easier to interpret from a figure with the total epsilon and topic coherence, with some fixed privacy budget split.
- The caption of Figure 5 says that both panels fix $\epsilon_2$ and vary $\epsilon_1$.
- Definition 4.1 uses the term "neighboring", while the following text uses the term "adjacent". Making these consistent would improve clarity.
- I think that "our statistic tend to exhibit normality" is a bit too strong given the 22% rejection rate in Table 4. Something like "our statistic is closer to normal than the alternatives" would be more appropriate.

Questions
- Is there a concrete privacy budget split that the experiments suggest? For example x% of total $\epsilon$ to DP LDA.
- Why did you choose the DP LDA algorithm of Zhu et al. (2016)?
- What is the confidence level for Table 4?

**Strengths And Weaknesses:**

The paper is mostly easy to read and understand, and introduces the related work well. The contributions themselves are very modest, but useful. The limitations of the work are also discussed well.

---

> ### Author Response · Authors · 2024-07-07
>
> Thank you for your questions and comments. We provide an response to your questions and a few comments below.
>
> - **Is there a concrete privacy budget split that the experiments suggest? For example x\% of total $\epsilon$ to DP LDA.** We chose to veer away from prescribing a definitive privacy budget split because as DP vocabulary selection or DP LDA algorithms improve or change based on implementation, we suspect that x\% prescribed for each will change. Additionally, prescribing a privacy budget split would shift the focus onto the specific DP mechanisms instead of the the modularity of the system proposed.
>
>  - **Why did you choose the DP LDA algorithm of Zhu et al. (2016)?** We chose the DP LDA algorthm from Zhu et al (2016) because it satisfies the same notion of adjacency and definition (approximate) of DP that our vocabulary selection method does. We include a brief note in the revisions in seciton 4.4 to highlight this decision.
>
> - **What is the confidence level for Table 4?** Because table 4 is data summary statistics, we assume that you are referencing Table 3. The significance level is .05. The updated copy contains a revised table caption indicating an $\alpha$ of .05 and we state the significance in the Appendix body at bottom of page 17.
>
>  - **Experimental results containing uncertainty estimates, at least for the tables.** We account for uncertainty by reporting the TPRs *across* multiple iterations. We determine the thresholds for each FPR using all observed MIA scores after running our tests through the dataset over many times with different data splits, shadow models, etc.
>
>  - **The performance of the full mechanism would be easier to interpret from a figure with the total epsilon and topic coherence, with some fixed privacy budget split.** We choose to omit this experimental result in the main paper body because the results are unsurprising and we think varying one $\epsilon$ while fixing the other highlights the individual effects of DP-LDA or DPSU. We argue that this set up provides stronger support for our claim that more of the privacy budget should be allocated to the DP learning algorithm, and the suggested experiment may clutter our results.
>
> Based on your comments we provided the following revisions. They appear in blue in the revised copy.
>
> - Fix the typo in the figure 5 caption.
> - Change the word choice in Definition 4.1 to include the term "adjacency" for consistency.
> - Change the our statement on normality in Appendix C to "Our analysis indicates that the distributions of $\tilde{\textbf{T}}_{in/out}(d)$ for our statistic tend to be normal more often than the alternatives."

---

> > ### Comment · Reviewer_NtZa · 2024-07-08
> >
> > Thank you for the response. Regarding the uncertainty estimates, just reporting the TPR as a single number is not an uncertainty estimate, even if you are computing it across multiple repeats. An uncertainty estimate would be something like a standard deviation estimate or a confidence interval, which you should be able to compute by bootstrapping over the repeats.

---

### Review · Reviewer_Ev4o · 2024-06-23

**Summary Of Contributions:**

The authors propose a membership inference attack against topic models that can confidently identify members of the training data in Latent Dirichlet Allocation models, and demonstrate the underlying privacy risks. They benchmark their method on 3 different datasets and compare against well motivated baselines in the literature. The results look quite impressive.

To mitigate these privacy vulnerabilities the authors propose to use differential privacy for topic modelling. They use known mechanisms for DP secure vocab selection and LDA training to obtain differentially private topic modelling. The authors conduct experiments to benchmark  performance of privacy attacks for different privacy budgets.

**Audience:**

Yes

**Claims And Evidence:**

Yes

**Requested Changes:**

Some notational/minor changes,

1. I presume $\phi$ in section 2.1 is a $V$ dimensional distribution. Please mention that.

2. Why is $D \in [0,\infty]^{M\times V}$?

3. ``We propose an $\ldots$ topic modeling provides.'' Missing "that"

4. Below equation (1), do you really mean to say that $\Phi$ is infeasible to calculate or do you mean $\mathbf T$?

5. For the experiment I presume that the threshold for Huang et al. was found by holding a fixed FPR?

**Strengths And Weaknesses:**

The paper is generally written well for the most parts, and the experimental results look quite impressive. A few questions and comments as follows,

1. The assumption that the adversary has access to the underlying data distribution, and that $\mathbf T$ is normally distributed is quite strong (as also corroborated by the results in Appendix C).
2. What is the $\mathcal G$ for the true model and the shadow models? Clearly that will have a huge impact on the performance of the attack. Alternatively, in the wild, $\mathcal G_{true}$ could be quite different from $\mathcal G_{shadow}$
3. The authors overly rely on the literature at places instead of explaining the intuition in their own words. For instance the authors cite Carlini et al. (and Mimno et al) extensively but please elaborate upon why

>  attacks that directly apply global thresholds do not consider document level differences on the learned model and fail to confidently identify members of the training data.

>  topic coherence can be a useful proxy for measuring the interpretability of topics.

and at other places.

4. The part on DP should be expanded especially i) why and how  DPSU and DP LDA works ii) what is "coherence" and why is it a good metric
5. It is difficult to understand the impact of different privacy budget $\epsilon$ as a number on its own as opposed to the performance on some downstream task. For instance is $\epsilon = 1$ at which the attack is almost useless too strong a privacy constraint to provide any utility for the downstream task?
6. Could you please elaborate upon why
> We do not expect performance to significantly decrease using disjoint datasets as observed by Carlini et al
7.  I am not sure I understand
>  The intuition behind our statistic is that a document is more likely generated by the target model when included the training data.

Isn't the intuition behind (3) the same as what has been used by Huang et al i.e. "document’s topic distribution tends to concentrate in a few topics when included (or duplicated) in the training data"

8.  (For my own understanding) What happens if we just use $\Phi^2$ (with perhaps some normalized version of \Phi)instead of $\max_{\theta} \theta * \Phi$ in equation 3?

---

> ### Author Response · Authors · 2024-07-07
>
> Thank you for taking the time to review our paper and your kind words. We respond your questions and comments in the order provided below.
>
> 1.  These adversarial assumptions are standard for many MIAs and somewhat realistic depending on model deployment/release. For example, a computational social scientist could disclose that the data from their study came from Twitter posts with a certain hashtag between two dates. In cases like this, an adversary would have complete access to the underlying data distribution.
>
> 2.  In our experiments, LDA defines the target and shadow models parameterized by $\mathcal{G}$. In the wild, most probabilistic topic model deployments are LDA or are some adaptation of LDA with minor changes in the generation processes. A mismatch in adversarial assumptions of $\mathcal{G}$ may impact attack accuracy, but we suspect to a lesser extent than a mismatch between target and shadow model in attacks against larger neural models. We edited the last paragraph of the limitations to comment on this point. There may also be differences in the learned model based on the data sampled which we acknowledge in point 6.
>
> 3. For brevity's sake we omit some explanation and rely on the existing literature. Our response to comment 4 reflects our changes.
>
> 4. In our paper revisions we provide more background on DPSU in section 4.2. We believe that the background for how DP LDA works is sufficiently covered by section 4.1. We added a sentence to section 4.4 to help explain topic coherence, but we'd also like highlight the limitations of automated topic model evaluation as noted in our Limitations section.
>
> 5. We measure utility directly with topic coherence for which we show performance across various settings of $\epsilon$. We assume that topic models with better topic coherence scores will perform better on downstream tasks.
>
> 6. Carlini et al show that attack performance does not significantly decrease using disjoint datasets because the shadow model training data is still drawn from the same underlying data distribution. In practice, the change in experimental set-up is related to the adversaries' accurately ability to sample from the underlying data distribution.
>
> 7. Our statistic uses a heuristic for the likelihood of the target document under the model. Empirically the likelihood of a document may increase because the "document’s topic distribution tends to concentrate in a few topics when included (or duplicated) in the training data," which our statistic can take advantage of while maintaining interpretability. To help make the difference more clear, we changed the first sentence of the 4th paragraph of section 3.3 to, "Compared to the statistics proposed by Huang et al, our statistic more directly exploits LDAs generative processes and is based on the observation that a document's likelihood under the target model increases when included in the training data."
>
> 8. When we take the max $\theta$, we are asking for the topic distribution that maximizes the likelihood of the document. Using a normalized version or scaled of $\Phi$ leaves a crucial part of the topic models document generation process out of the equation. We experimented with many candidate solutions and those that incorporated some simple transformation of $\Phi$ (even when $\Phi$ is reduced only to represent words in the target document) tended to perform poorly. In short, we do not get enough information about the target documents affect on the model with these types of statistics.
>
> Based on your requested changes, we respond to and edit the paper in the following places. All changes are highlighted in blue.
>
> - We change the second sentence in section 2.1 to "LDA assumes that each document $d \in D$ is represented by a $k$-dimensional document-topic distribution $\theta \sim \text{Dirichlet}(\alpha)$, and the entire corpus is represented by $k$ $V$-dimensional topic-word distributions $\phi \sim \text{Dirichlet}(\beta)$."
>
> - This references the bag-of-words approach where all $M$ documents are represented as a $V$ dimensional vector with entries that correspond to each words' document-word count. It does not add much to section 2.0 so we just omit the notation.
>
> - We believe that intractable may be a better term. We changed this sentence to "However, the ratio in Equation 1 is intractable because it requires integrating over all $\Phi$ with all possible $D$ with and without $d$."
>
> - Yes. To clairfy, we change the last sentence to of section 3.5 to "We empirically estimate our attacks' performance by evaluating the TPR at all FPRs and plot the attack's ROC curve on log-scaled axes."

---

### Decision · Action_Editor_hLWW · 2024-07-30

**Recommendation:** Accept as is

**Comment:**

All reviewers are happy with the author responses and recommend acceptance without further revisions.

The paper looks mostly publishable to me, but I would ask the authors to check the references for the camera ready: some are missing publication details such as conference names and many proper names (e.g. Rényi, Dirichlet) in lowercase.

**Audience:**

All reviewers agree that a part of TMLR audience would be interested in findings of the paper.

**Claims And Evidence:**

All reviewers agree that the submission is supported by accurate, convincing and clear evidence.